# Chagas Disease Diagnosis with *Trypanosoma cruzi*-Exclusive Epitopes in GFP

**DOI:** 10.3390/vaccines12091029

**Published:** 2024-09-08

**Authors:** Andressa da M. Durans, Paloma Napoleão-Pêgo, Flavia C. G. Reis, Evandro R. Dias, Luciana E. S. F. Machado, Guilherme C. Lechuga, Angela C. V. Junqueira, Salvatore G. De-Simone, David W. Provance

**Affiliations:** 1Center for Technological Development in Health, National Institute of Science and Technology for Innovation in Neglected Population Diseases, Oswaldo Cruz Foundation, Rio de Janeiro 21040-900, Brazil; andressa.durans@fiocruz.br (A.d.M.D.); paloma.pego@fiocruz.br (P.N.-P.); flavia.reis@fiocruz.br (F.C.G.R.); evandrodias@aluno.fiocruz.br (E.R.D.); lmachado@usp.br (L.E.S.F.M.); guilherme.curty@fiocruz.br (G.C.L.); salvatore.simone@fiocruz.br (S.G.D.-S.); 2Epidemiology and Molecular Systematics Laboratory, Oswaldo Cruz Institute, Oswaldo Cruz Foundation, Rio de Janeiro 21040-900, Brazil; 3Interdisciplinary Laboratory of Medical Researchers, Oswaldo Cruz Institute, Oswaldo Cruz Foundation, Rio de Janeiro 21040-900, Brazil; 4Laboratory on Parasitic Diseases, Oswaldo Cruz Institute, Oswaldo Cruz Foundation, Rio de Janeiro 21040-900, Brazil; junqueir@ioc.fiocruz.br; 5Department of Genetics and Biology Evolution, Institute of Biosciences, University of São Paulo, São Paulo 05508-090, Brazil; 6Program of Post-Graduation on Science and Biotechnology, Department of Molecular and Cellular Biology, Biology Institute, Federal Fluminense University, Niterói 22040-036, Brazil

**Keywords:** Chagas disease, Green Fluorescent Protein, Serodiagnosis, ELISA, *Trypanosoma cruzi*

## Abstract

Serological tests are critical tools in the fight against infectious disease. They detect antibodies produced during an adaptive immune response against a pathogen with an immunological reagent, whose antibody binding characteristics define the specificity and sensitivity of the assay. While pathogen proteins have conveniently served as reagents, their performance is limited by the natural grouping of specific and non-specific antibody binding sites, epitopes. An attractive solution is to build synthetic proteins that only contains pathogen-specific epitopes, which could theoretically reach 100% specificity. However, the genesis of de novo proteins remains a challenge. To address the uncertainty of producing a synthetic protein, we have repurposed the beta barrel of fluorescent proteins into a receptacle that can receive several epitope sequences without compromising its ability to be expressed. Here, two versions of a multiepitope protein were built using the receptacle that differ by their grouping of epitopes specific to the parasite *Trypanosoma cruzi*, the causative agent for Chagas disease. An evaluation of their performance as the capture reagent in ELISAs showed near-complete agreement with recommended diagnostic protocols. The results suggest that a single assay could be developed for the diagnosis of Chagas disease and that this approach could be applied to other diseases.

## 1. Introduction

First described over a century ago [1], American trypanosomiasis is caused by infections of the protozoan parasite *Trypanosoma cruzi*. More commonly referred to as Chagas disease (CD), it is one of 20 tropical neglected diseases declared by the WHO and continues to be a major health concern [2]. An estimated 6–7 million people, primarily throughout twenty-one countries of Latin America, are infected, with up to 75 million people at risk of contracting the disease [3]. Within endemic regions, the primary mode of transmission is through bloodsucking insects from the family triatomines [4]. Other modes of transmission include non-pasteurized food products, blood transfusions, organ transplants, and mother-to-fetus transfer [5]. Globally, the incidence of *T. cruzi* infections is increasing geographically as a result of emigration from endemic regions to non-endemic regions [6,7].

The period immediately after an infection, the acute phase, is generally asymptomatic [8]. Due to the high levels of circulating parasites, the detection of the sanguineous trypomastigote form through the microscopic examination of blood smears is sufficiently sensitive to confirm infection and can also differentiate *T. cruzi* from *Trypanosoma rangeli* [9]. A loss of detectable, circulating parasites indicates that an infection has entered an indeterminate, chronic phase that can span multiple decades with no overt symptoms [10]. After this transition, direct parasitological methods for diagnosis are unreliable and elevate the detection of antibodies, principally IgG, against *T. cruzi* antigens via serological methods as more appropriate [11].

As around one-third of Chagasic patients will present with life-threatening cardiac and/or digestive symptoms [12,13,14], there is a clear need to identify infected individuals. Several immunological tests based on different technological platforms are available for CD diagnosis including indirect hemagglutination assays [15], indirect immunofluorescence assays [16,17], enzyme-linked immunosorbent assays (ELISA) [18,19], and chemiluminescent assays [20]. Still, no ideal serological test exists to provide the desired performance across all endemic areas of the disease that account for the genome diversity between the six discrete typing units of *T. cruzi* (TcI–TcVI) and variations in serological titers, which can be low in patients in some regions such as the Amazon of Brazil [21]. Currently, the WHO and the Brazilian Ministry of Health recommend the use of two independent serological tests to reach a conclusive diagnosis [22,23]. In the case of ambiguous or discordant results, a third technique is suggested using a new patient sample.

The nonexistence of a gold standard assay increases the time and cost of diagnosis, which limits the detection of chronically infected individuals. Since the performance of an assay reflects the antibody binding characteristics of the reagent employed, the problem can be isolated to the antigens used to detect anti-*T. cruzi* antibodies. While recombinant DNA techniques have expanded our ability to prepare different antigens for use in the immunological diagnosis of CD [24,25], major difficulties remain with regard to sensitivity and cross-reactivity, especially with other protozoan parasites such as *Leishmania spp.* and *Trypanosoma rangeli* [26,27].

One issue is the presence of amino acid sequences in the capture reagent that are not exclusive to *T. cruzi* and can bind antibodies, which is a near-certainty in natural proteins. The solution is to only use qualified epitopes, which represent the smallest unit bound by an antibody [28]. While a large number of epitopes have been identified for *T. cruzi*, no approach has been described for consistently building multiepitope proteins. As the need for a high-performance capture reagent is not restricted to CD, we sought to develop a systematic method that could be easily implemented for the design of new proteins. To that end, we recognized that the ß-barrel structure of fluorescent proteins could serve as the core structure of synthetic proteins to receive exogenous sequences in the regions between the staves of the barrel. Here, we present two chimeric proteins that combine ten linear B-cell epitopes from *T. cruzi* into a single open reading frame and show its potential to serve as an immunological mimic for the specific and sensitive detection of reactive anti-*T. cruzi* patient IgG antibodies for the accurate diagnosis of CD.

## 2. Materials and Methods

### 2.1. Ethical Considerations for the Use of Human Serum

This investigation was approved by the Ethical Committee of the Oswaldo Cruz Institute, Oswaldo Cruz Foundation, Rio de Janeiro (RJ) (CAEE: 52892216.8.0000.5248), which included the registration of collaborations with Central Public Health Laboratories (LACEN) from four Brazilian states (Ceará, Maranhão, Paraíba, and Sergipe) and the study led by Dr. Angela Junqueira that collected samples in the field from individuals in Barcelos municipality in Amazonas state (CAAE: 79922017.8.0000.5248).

### 2.2. Serological Samples

Serological samples obtained from Central Public Health Laboratories (LACEN) from Brazil (Ceará, Maranhão, Sergipe and Paraíba) included Chagas chronic disease (n = 61), samples negative for Chagas disease (n = 52), visceral leishmaniasis (n = 32), and dengue virus (n = 20). All samples were retested for chronic Chagas disease using two commercial serological kits: IFI-Chagas (BioManguinhos, Rio de Janeiro, Brazil) and Chagastest ELISA Rec. v3.0 (Wiener Lab, São Paulo, Brazil). Patient sera from Barcelos, AM, Brazil, were tested using commercial serological kits (IIF and ELISA) to identify positive (n = 60) and negative individuals (n = 75). Only samples with concordant results for both serologies were used in the ROC curve and reactivity index analysis. In addition, serological samples from people without infection were used at the time of collection (Healthy) from blood donor bank HEMORIO, Rio de Janeiro, Brazil (n = 24). For the analysis of analytical sensitivity, the lyophilized pools NIBSC 09/188 (TCI) and NIBSC 09/186 (TCII) from the World Health Organization (WHO; Geneva, Switzerland, were obtained from the collaborating center at the National Institute for Biological Standards and Controls (NIBSC) (Hertfordshire, UK) and used as recommended in a 2-fold dilution series from 1:2 to 1:256.

### 2.3. Cloning, Expression, and Purification

All molecular biology and recombinant protein procedures followed institutional biosecurity requirements for genetically modified organisms (Comissão Interna de Biossegurança IOC/Fiocruz protocol No°5/5342017). The coding sequence of DxCruziV1 was synthesized as a linear DNA fragment without the first or last epitopes through GeneART (ThermoFisher, Waltham, MA, USA) with a 5′ NdeI site and a 3′ XhoI site for cloning into pET-28a (+) (Novagen, Darmstadt, Germany). The primer dimers TAAATTCGCGGAACTGCTGGAACAGCAGAAAAACGCGCAGTTCCCGGGTAAAGCTAG and CTA GCTAGCTTTACCCGGGAACTGCGCGTTTTTCTGCTGTTCCAGCAGTTCCGCGAATT were used to introduce the first epitope between the NdeI and NheI sites. The last epitope was added by the primer dimers AATTCAAACAGAAAC GTGCGGCGGAAGCGACCAAATGAC and TCGAGTCATTTGGTCGCTTCCGCCG CACGTTTCTGTTTG into the EcoRI and XhoI sites. The coding sequence of DxCruziV2 was synthesized as a gBlock (Integrated DNA Technologies, Coralville, IA, USA) with NdeI XhoI sites for cloning into pET-28a (+). After confirmation via sequencing, *E. coli* BL21(DE3) bacteria were transformed for expression through the T7 system [29,30,31]. Ten colonies were screened for protein expression at a small scale (5 mL), and the highest-expressing colony was chosen for a 100 mL culture to produce protein via purification. Following a 3h induction period with IPTG (1 mM), bacteria were collected by centrifugation (3500× *g*, 20 min @ 25 °C). The pellet was resuspended in lysis buffer (50 mM Tris-HCl, pH 8.0; 100 mM NaCl), followed by sonification on ice (3 min, amplitude 20, pulse of 30 s and pause 60 s). Inclusion bodies were collected and washed three times with lysis buffer plus detergent (1% Triton X-100) via centrifugation (16,000× *g*, 20 min @ 4 °C). The final pellet was solubilized in Tris buffer (50 mM Tris-HCl, pH 8.0 containing 100 mM NaCl) with 4M urea. After clarification (16,000× *g*, 20 min @ 4 °C), the sample was applied to a HiTrap^®^ HP column (GE Healthcare Life Sci, Piscataway, NJ, USA) on an Äkta 10 chromatographic system. Following 3 columns of Tris buffer with 4 M urea, urea was omitted for the next 3 column volumes, followed by 3 column volumes of Tris buffer with 25 mM imidazole. Bound DxCruziV1 or DxCruziV2 was eluted by a gradient of imidazole up to 500 mM. Elution fractions were analyzed via SDS-PAGE.

### 2.4. Enzyme-Linked Immunosorbent Assays

In-house ELISA assays began by coating 96-well plates (H2B-High Binding, Corning^®^, Sigma-Aldrich, Burlington, MA, USA) with 500 ng/well of either DxCruziV1 or DxCruziV2 in sodium carbonate-bicarbonate buffer (0.05 M, pH 9.6) with an overnight incubation at 4 °C. After washing 3x with PBS, plates were blocked with 1x PBS-T (pH 7.4 with 0.05% Tween^®^ 20) with 5% powdered non-fat milk at 37 °C. After 1h, buffer was removed, and patient serum diluted 1:250 in 50 µL of PBS-T was added for another 1h of incubation at 37 °C before washing 3× with PBS-T. For detection with alkaline phosphatase (AP), goat anti-human IgG (H+L) antibody labeled with AP (Material # 5220-0303; KPL-SERACARE, Milford, MA, USA) was diluted 1:5000 in PBS-T, incubated 1h at room temp and washed three times with PBS-T, buffer was removed, and 1-Step™ PNPP Substrate Solution (Thermo Fisher, Scientific Inc., Rockford, IL, USA) was added, followed by incubation at room temperature in the dark for 30 min. Enzymatic activity was stopped by the addition 1 M NaOH, and the absorbance was measured at 405 nm in an automated plate reader (Model 680; BioRad, Hercules, CA, USA).

In assays employing the serial dilution of the WHO International Standards [32] [IS 09/188 and IS 09/186], ELISAs were performed with the AP-labeled secondary antibody as above or with a horseradish peroxidase (HRP) detection system that employed a goat anti-human IgG Fc-specific antibody conjugated to HRP (Cat # A0170, Sigma-Aldrich, St. Louis, MO, USA) diluted 1:60,000. Reactivity was revealed by the addition of 1-Step™ TMB Ultra (Scienco, SC, Brazil). After a 15 min incubation period at room temperature in the dark, the reaction with stopped with 0.5 M H_2_SO_4_ and the absorbance measured at 450 nm. Raw data were exported to Excel V16.0 (Microsoft, Seattle, WA, USA) to subtract the background for graphing in Prism V10 (GraphPad Software, San Diego, CA, USA).

### 2.5. Spot Synthesis Analysis

The coding sequence of DxCruziV1 was converted into a library of consecutive 14 amino acid peptides with an 8-residue overlap that was synthesized onto an Amino-PEG_500_-UC540 cellulose membrane using an Auto-Spot Robot ASP-222 (Intavis Bioanalytical Instruments AG, Köln, Germany) according to a previous SPOT synthesis protocol [33]. The peptide IHLVNNESSEVIVHK from *Clostridium tetani* was used as a positive control. A spot without peptide served as a negative control. During synthesis, acetylation with acetic anhydride (4%, *v*/*v*) in N,N-dimethylformamide followed each coupling reaction. Nascent peptides were rendered reactive through the removal of the Fmoc protective group with piperidine. After the last cycle, amino acid side chains were deprotected using dichloromethane–trifluoracetic acid–triisobutylsilane (1:1:0.05, *v*/*v*/*v*), washed with methanol, and hydrated with TBS (50 mM Tris-buffer saline, pH 7.0). Membranes were blocked overnight at 4 °C with agitation with TBS-CT (Tris-buffer saline, 3% casein, 0.1% Tween^®^ 20, pH 7.0). After five washes in TBS-T (Tris-buffer saline, 0.1% Tween^®^ 20, pH 7.0), the library was exposed for 2 h to a pool of ten Chagas-disease-positive or -negative sera diluted 1:200. Membranes were washed three times in TBS-T before a 1h incubation period at 37 °C with alkaline-phosphatase-labeled rabbit anti-human IgG (1:5000 in TBS-CT). Next, the membrane was washed extensively in TBS-T and exchanged into CBS (50 mM citrate-buffer saline, pH 7.0), and reactive peptides were revealed via chemiluminescence using CDP-Star^®^ Substrate (Cytiva, Marlborough, MA, USA) with Nitro-Block-II™ Enhancer (PerkinElmer Life Sciences, Boston, MA, USA). Signals were captured by an Odyssey CLx (LI-COR, Lincoln, NE, USA), and the intensities were quantified with TotalLab TL100 (v2006, Nonlinear Dynamics Ltd., Newcastle upon Tyne, UK). Intensity was normalized by defining the highest signal as 100%.

### 2.6. Statistics

Graphical and statistical analyses were performed using Prism V10 software. Statistical differences were identified using the *t*-test. Significance was considered if the *p*-value was less than 0.05. The cutoff values were obtained from a receiver operating characteristic (ROC) analysis of the data, and the area under the receiver operating characteristic curves was used to estimate diagnostic accuracy. The Kappa index represented the agreement between the diagnoses reported by the LACENs and the in-house ELISAs. The reactivity index (RI) was calculated by dividing the absorbance by the cutoff value. A result equal to or greater than 1.1 was considered reactive, while a result of less that 0.9 was considered non-reactive. Values between 0.9 and 1.09 were classified as borderline and inconclusive.

## 3. Results

### 3.1. Design and Production of DxCruziV1

The first obstacle to developing a new immunological reagent for diagnosing Chagas disease was the need for 100% specificity to minimize the potential for false-positive results. By restricting the pathogenic components to linear B-cell epitopes that have been empirically determined over decades to be specific to *T. cruzi* (Table 1), it was possible to exclude non-specific regions. Since epitopes are generally nonstructured segments on the surfaces of proteins, we next searched for a natural protein structure to serve as the core/interior of chimeric proteins that could be highly modified at multiple sites and would be expected to produce a usable protein. Several studies described the insertion of extraneous amino acid sequences within various regions of fluorescent proteins, all outside of the beta-barrel structure (Appendix A). Further examination showed that many amino acid changes that altered the biochemical properties of green fluorescent protein during its directed evolution to a more stable protein were within the ß-strands comprising the ß-barrel (Appendix A). A combination of the amino acid changes described was chosen to form the ß-receptacle.

To develop a platform for building multiepitope proteins, the back translation to nucleotides was examined for silent changes to generate unique restriction sites in the loop regions’ intervening ß-strands. The addition of single amino acids was also investigated for the introduction of restriction sites. A set of 12 unique restriction sites was identified that could define ten insertion points in the ß-receptacle (Figure 1A), which allow for the insertion or exchange of epitope sequences through common molecular biology methods. The final sequence of the first multiepitope protein, DxCruziV1, was obtained via the in silico combination of the epitopes with the ß-receptacle, which generated duplications of amino acids in the restriction sites (Figure 1B). 

### 3.2. Design and Production of DxCruziV1

Before DNA synthesis, the predicted structure was determined through the I-Tasser server [47,48]. Figure 2A shows three angles of the model with the epitopes in red and the ß-receptacle in green. The retention of the ß-barrel structure suggested that the addition of the epitopes did not compromise the final structure, which was confirmed by the successful expression of DxCruziV1 in non-fluorescent inclusion bodies (Appendix A). Protein solubilized in urea could be exchanged into physiological buffer on a column during purification.

### 3.3. Conditions for Performing In-House ELISAs

ELISAs were optimized by examining the minimum quantity of DxCruziV1 necessary to sensibilize 96-well plates for detecting antibodies in the two WHO International Biological Standards for Chagas disease (Appendix A). As there was a loss in signal between the assays for the 1:64 dilution of IS 09-186, 500 ng of DxCruziV1 was chosen. The patient serum dilution factor was chosen based on the optical densities measured from a panel of four high- and four low-titer samples (Appendix A). As a higher dilution factor could minimize cross-reactivity, and there was a loss in the signal going to the 1:500 dilution factor for the samples with the lowest titers, a dilution factor of 1:250 was chosen. 

To evaluate the performance of DxCruziV1 in an ELISA format, a panel of sera from one-hundred and thirty-five (135) patients was obtained from areas covered by IS 09-186 (Appendix A), representing individuals with some of the lowest titers of antibodies. Figure 2B shows the absorbance values measured for the group of patient samples previously diagnosed according to the recommended guidelines of two independent assays to be infected with *T. cruzi* (green diamonds; n = 60) and not infected (red circles; n = 75). From a receiver operating characteristic analysis (Appendix A), a cutoff value of 0.3170 displayed 100% sensitivity and specificity with a perfect Kappa index value of 1 and a likelihood ratio of 76.0 (Table 2). The cutoff was used to calculate the reactivity index from control samples for non-specific binding and cross-reactivity (Figure 2C). Only a single patient sample from the group diagnosed with malaria displayed a reactivity greater than one.

### 3.4. Epitope Contribution to Signal

While the performance of DxCruziV1 was exceptional, we wanted to determine which of the epitopes contributed to the capture of antibodies. The technique of spot synthesis analysis was applied to examine the antibody binding characteristics of the whole protein. An array of 14 amino acid peptides offset by six amino acids and representing the coding sequence were synthesized in place on a cellulose membrane, followed by incubation with a pool of sera from Chagasic patients or non-Chagasic patients. Figure 3A shows the image from the chemiluminescent signals captured for binding human IgG antibodies in the positive samples (rows A–C) and negative samples (rows D–F). A semiquantitative plot of the signals, normalized by the reactivity of the positive control for 100% and negative controls for 0%, shows that several, but not all, epitopes were recognized by antibodies in the pool of patient sera (Figure 3B, top graph). In the plot of the reactivity against non-affected individuals (Figure 3B, bottom graph), no regions of DxCruziV1 displayed values of >30%, which is our cutoff for reactivity. Ultimately, epitopes 2, 5, and 7, which were included in the design of DxCruziV1, did not appear to be contributing to the capture of anti-*T. cruzi* antibodies (Table 1).

### 3.5. Design, Production, and Performance of DxCruziV2

From the spot synthesis analysis results, a new multiepitope protein was designed that replaced the non-reactive regions with new epitopes (Table 1). In addition, epitope 2 (KAAAAPA) was placed in the carboxy terminus before epitope 10 (KQRAAEATK). Due to the size of epitope 13, PPSGTENNKPATG, the sequence GSGTSWKGS was included to minimize the potential for introducing a change into its structure or that of the ß-barrel. As with DxCruziV1, the computer modeling of DxCruziV2 suggested that the ß-barrel structure would be retained, and a protein could be produced (Figure 4A). 

Following DNA synthesis, the new design was successfully expressed in bacteria using the same protocol as DxCruziV1. For a direct comparison to the performance of DxCruziV1, all the same conditions were employed for the in-house ELISAs, including patient samples. Figure 4B shows each individual’s optical density measured at 405 nm. A receiver operating characteristic analysis returned two cutoff values of 0.26 and 0.3065 that displayed equivalent sensitivities (96.67%) with specificities of 98.68% and 100%, respectively (Table 2). The positive predictive value at both cutoffs was 98.3, while the negative predictive values were 97.4 and 96.1 at the lower and higher cutoffs. Applying the cutoff value most similar to DxCruziV1 (0.31), the reactivity index was calculated for the control samples for non-specific binding and cross-reactivity (Figure 4C). Only a single patient sample from the group diagnosed with visceral leishmaniosis displayed a reactivity greater than one, with two others being borderline (RI 1 ± 0.1). 

### 3.6. Analytical Sensitivity of DxCruziV1 and DxCruziV2

Considering that the DxCruziV1 and DxCruziV2 proteins were generated and evaluated years apart, their analytical sensitivity was analyzed at the same time using the International Biological Standards from the WHO. Two detection systems, alkaline phosphatase (AP) and horseradish peroxidase (HRP), also were examined. Figure 5 shows the results from a 1:2 serial dilution of IS 09/188 (Figure 5A) and IS 09/186 (Figure 5B). Based on the cutoffs determined previously, both multiepitope proteins, in combination with an HRP detection system, meet or exceed the recommended maximum dilution from the WHO of 1/64 by registering a reactivity index (RI) greater than 1. Only DxCruziV1 with the AP system displayed an RI of less than one at 1/64. Maximum analytical sensitivity was shown by DxCruziV2 using an HRP labeled as secondary at a dilution of 1/254 for both standards, which was four-fold greater than recommended.

The difference in performance between the two biological standards suggested that each multiepitope protein could display different sensitivities concerning the geographical origins of the patient samples. As our samples traverse the northern region of Brazil, from the Amazon to the East Coast, we performed Brown–Forsythe and Welch ANOVA tests without assuming equivalent standard deviations. For DxCruziV1, no differences in performance related to the geographical origins of patient samples were calculated. In comparing the performances of DxCruziV1 and DxCruziV2, Figure 5C shows a notable difference in all states except Maranhão. DxCruziV2 showed significantly greater reactivity than DxCruziV1 in Amazonas and Sergipe in Brazil. Across the different locations, DxCruziV2 showed substantial differences in performance only in comparison to samples from Maranhão (Figure 5D).

## 4. Discussion

There remains an unmet need for the detection of chronic CD through a singular serodiagnostic test that delivers high-confidence, actionable results. Treatment for CD is more effective when initiated most proximal to the start of an infection [50]. Furthermore, an economic model estimates that early identification and treatment would be highly cost-effective [51]. Currently, the approved diagnosis for chronic CD is the detection of anti-*T. cruzi* antibodies in two different serological tests [22]. In cases of divergence, a third confirmation test is required. These requirements reflect the absence of a gold-standard assay that can definitively detect anti-*T. cruzi* antibodies, which contributes to higher costs and longer execution times, as well as an underdiagnosis of CD that precludes treatment. 

The performance of a serological test is most directly related to the antibody binding characteristics of the defining capture reagent. At a higher resolution, it is the combined reactivity for all of the antibody binding sites, epitopes, contained within the reagent. For specificity, the epitopes should be unique to the pathogen, while their quantity relates to sensitivity. The search for biomarkers in *T. cruzi* for diagnosis has a long, storied history, and many have been incorporated, in various forms, into currently available commercial kits. However, their performances across different endemic regions suggest that the composition of their capture reagents contains an insufficient number of antibody binding sites with reactive, non-specific regions that can return false negative and positive results, respectively [49,52].

Here, we began with many of the same biomarkers but were restricted to the smallest reported unit for the epitope (Table 1). To combine the ten epitopes into a single synthetic protein, we considered the capacity of the ß-barrel from fluorescent proteins to serve as the core structure. Multiple lines of directed evolution have shown that many of the biochemical characteristics, such as the folding rate and stability, which are also favorable for immunological reagents, are delivered by the specific sequences in the ß-sheets [53,54,55,56]. Moreover, the connecting loop sequences between the eleven staves can be increased or replaced [57,58,59,60,61,62,63,64,65], and the lengths of some are nearly equal to those of epitopes. Lastly, the projected position of the epitopes would be expected to permit their interaction with antibodies.

While multiple examples exist for the insertion of sequences from 20 to 50 amino acids into 1, 2 or 3 loops with the retention of fluorescence, the maximum reported number of insertion sites was 3 to 6 loops of GFP to form fluorobodies [66]. The apparent intention to retain fluorescence may have restricted experiments on the insertion of sequences into more loops simultaneously. For our intended applications, fluorescence was not essential. Our primary need was to establish an approach for reliably producing proteins containing a high 7number of exogenous sequences, which is a major unknown for protein design [67,68]. The first deca-epitope protein design, DxCruziV1, proved to be easily expressed in bacteria. Moreover, it displayed 100% sensitivity and 100% specificity, which has been an elusive achievement for immunological reagents to diagnose Chagas disease. 

By performing a spot synthesis analysis [69], it was possible to observe the effective contribution of each linear epitope to the production of a signal. Three sequences showed low reactivity as peptides, and each was a variation in another epitope included in DxCruziV1. Two contained a single amino acid difference and the other was a shift in the repeated sequence. These were replaced to generate the second version of the chimeric protein, DxCruziV2. It also differed due to the addition of a balancer sequence (GSGTSWKGS) to epitope 13 and the transfer of epitope 3 to the carboxy terminus. Although its absorbance values appeared to be greater than those for DxCruziV1, the performance of DxCruziV2 showed a lower sensitivity (96.67%). As this could involve a deterioration of patient samples stored in the freezer over the years between their evaluations, their performances were directly compared using the WHO International Standards for Chagas disease. The analytical sensitivity of DxCruziV2 is clearly higher than that of DxCruziV1 and easily meets the recommended target dilution factor of 1:64 [32]. In combination with HRP, DxCruziV2 showed sensitivity up to a dilution of 1:256. In an evaluation of commercial kits approved by ANVISA, the Brazilian Federal regulatory agency for human medical applications, no assay met the target dilution for IS 09/188, and only one was sufficiently sensitive for 09/186 [49].

Additional studies with the same commercial tests showed significant differences in their performances in relation to the geographical origins of patient samples [52], which was also suggested for our proteins by the results obtained with the two international standards that represent different regions of the Americas. Comparing their performances, DxCruziV2 showed a significant improvement for most regions, suggesting that the epitopes chosen as replacements were reactive. The exception was with samples from Maranhão. It is the only region where DxCruziV2 showed a significantly lower performance in comparison to its performances in other regions. A statistical analysis of DxCruziV1 did not show any significant geographical differences in its performance. This observation suggests that the group of epitopes chosen for DxCruziV2 was not fully optimized for this region.

A BLAST analysis of the amino acid sequences showed several with a high level of homology, up to 100%, to sequences in other pathogens (Appendix A). Theoretically, this would suggest a potential for cross-reactivity. As an example, the epitope from the KMP11 protein has 100% identity against *T. rangeli* and *T. brucei gambiensi*, as well as 94.12% against *L. braziliensis*, *T. rangeli*, and *Crithidia* sp. While the whole protein was recognized by Leishmaniasis patient samples, no reactivity was observed for the two linear epitopes mapped [34]. This suggests that there are functional differences in antigen processing that influence the use of a particular sequence as an epitope and the generation of an antibody that only experimentation can definitively determine.

This highlights a major distinction of our approach for building synthetic proteins onto a ß-barrel backbone. As the core tertiary structure of the final proteins is not expected to deviate significantly, which can be corroborated by computer modeling, the choice of which epitopes to incorporate becomes paramount. This epitope-centric approach to serodiagnostics focuses the development of a new reagent on defining the desired performance first. This constrains the search for epitopes that meet those qualifications through the use of sera that reflect the target performance profile. As the system is best aligned to linear B-cell epitopes, numerous high-throughput techniques are available including peptide arrays and phage display for application to other pathogens.

## 5. Conclusions

High-performance immunological reagents for the diagnosis of chronic Chagas disease have been generated through the incorporation of ten *T. cruzi*-specific epitopes into the loops of the ß-barrel structures of fluorescent proteins. The repurposing of the ß-barrel into a receptacle for epitope sequences resolves multiple issues that plague the field of serodiagnosis. The ability to designate the antibody binding regions present within a designer protein eliminates the reliance on natural proteins and their suboptimal grouping of epitopes that diminish specificity and sensitivity. The choice of highly reactive and specific epitopes can deliver near-perfect performance, while the ß-barrel structure contributes to the formation of viable proteins. As there are no anticipated restrictions on fields to which epitopes can be employed, this approach can be applied to a wide range of applications. 

## 6. Patents

DWPJ, AMD, PNP, and SGS are listed as inventors on patent submission protected listings in Brazil (BR10.2019.017792.6), the USA (PCT/BR2020/050341), Europe (PCT: 26 June 2023), India (PCT: 26 June 2023), and China (PCT: 26 March 2023). Provisional patents were filed by FIOCRUZ. The protein receptacle, the method for receptacle production, and the antigenic peptide sequences described in this study may serve as future sources of funding. The funding agencies had no role in study design, data collection, data analysis, publication decisions, or manuscript preparation.

## Figures and Tables

**Figure 1 vaccines-12-01029-f001:**
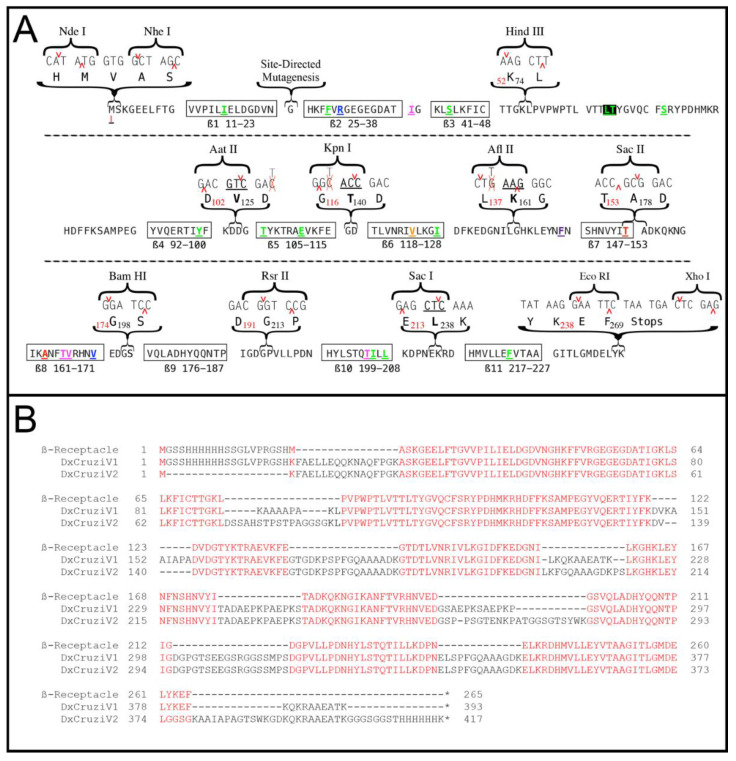
Sequences of the ß-receptacle, DxCruziV1, and DxCruziV2. Panel (**A**) shows the amino acid sequence selected as the ß-receptacle along with a back translation to its DNA sequence. For insertion into pET28(a), a 5′ NdeI site and a 3′ XhoI site were used to remove the multiple cloning sites. Based on the sites previously used to insert extraneous sequences (Appendix A), codons were chosen to generate unique restriction sites for HindIII, SacII, BamHI, and RsrII. The addition of a single amino acid (bold beneath the underlined codon) could form the unique sites AatII, KpnI, AflII, and SacI, which did not eliminate fluorescence. Colored amino acids depict amino acids altered via the directed evolution of eGFP (Appendix A). Panel (**B**) shows the protein sequences of the ß-receptacle, DxCruziV1, and DxCruziV2 aligned using Muscle in SnapGene (V7.0.2, GSL Biotech, San Diego, CA, USA). Conserved amino acids are shown in red.

**Figure 2 vaccines-12-01029-f002:**
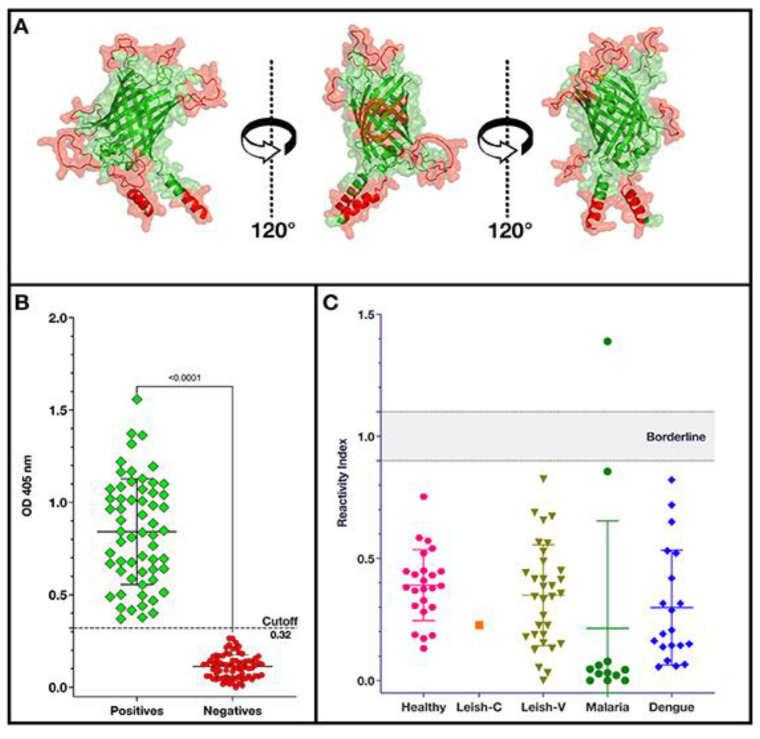
Projected structure of DxCruziV1 and its performance as an antibody capture molecule for in-house ELISAs. Panel (**A**) depicts three views of the tertiary structure predicted on the I-Tasser server with the inserted epitopes in red and the core structure in green. Panel (**B**) shows the optical density measured at 405 nm for patient samples previously diagnosed by the LACENs in the states of Maranhão, Sergipe, Ceará, and Paraíba IN Brazil to be positive (green diamonds; n = 60) and negative (red dots; n = 75) for Chagas disease. Panel (**C**) shows the reactivity indexes of healthy individuals (n = 24) and patients with inactive cutaneous Leishmaniasis (Leish-C; n = 1), visceral Leishmaniasis (Leish-V; n = 32), malaria (n = 12), or dengue (n = 20) calculated with the cutoff value obtained from an ROC analysis of the data in Panel (**B**).

**Figure 3 vaccines-12-01029-f003:**
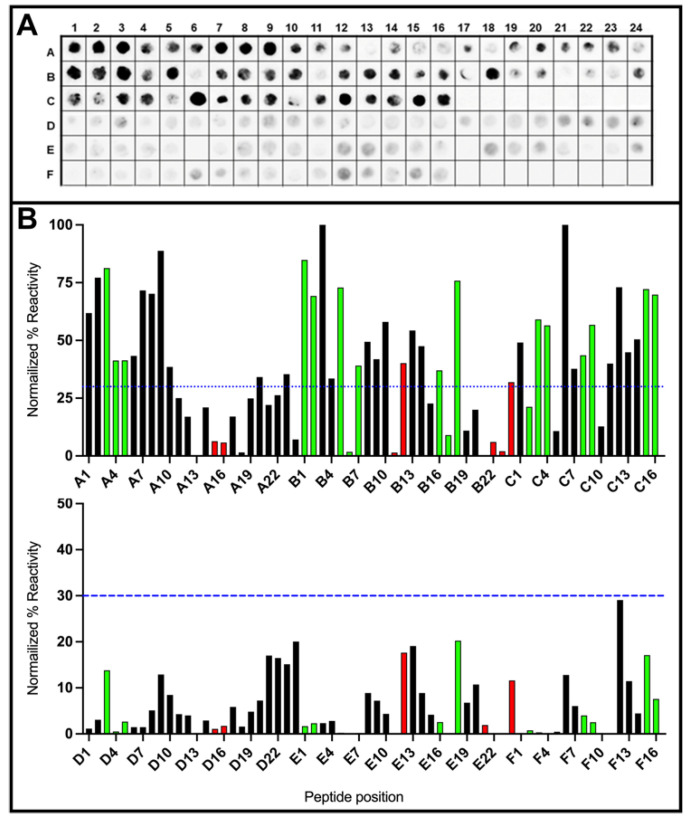
Spot synthesis analysis of DxCruziV1. The coding sequence of DxCruziV1 was converted into a library of 64 consecutive peptides of 14 residues with a 6-amino-acid overlap that was synthesized in duplicate directly onto a cellulose membrane as a grid of 3 rows (A–C or D–F) and 24 columns. Panel (**A**) depicts the chemiluminescent image of the membrane after incubation with a pool of sera (n = 10) from Chagasic patients (Rows A–C) or non-Chagasic individuals (Rows D–F) for the detection of human IgG antibodies. Panel (**B**) displays the respective quantification of the signals measured from the membrane in Panel (**A**). The blue dotted lines indicate the cutoff point used to select positives from negatives. Below the line indicates background. The list of peptides and their positions is presented in Appendix A.

**Figure 4 vaccines-12-01029-f004:**
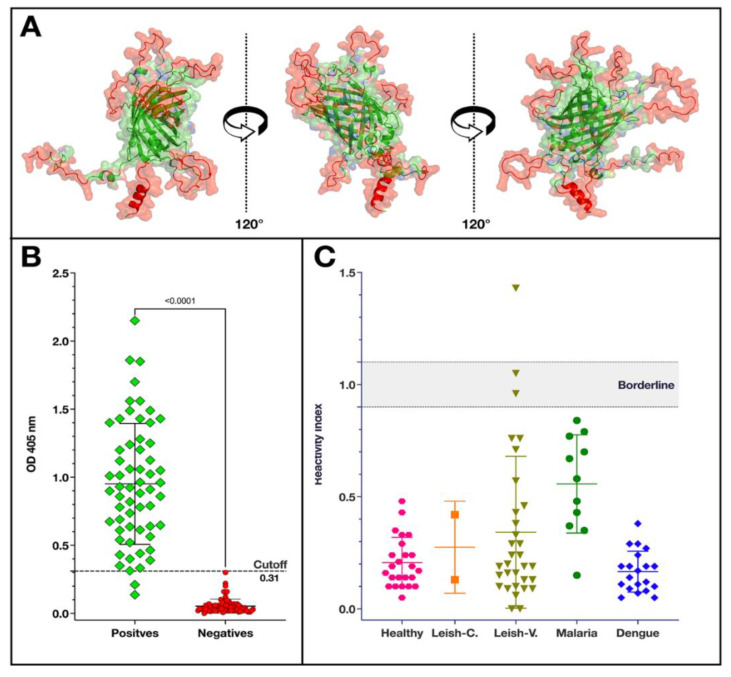
Projected structure of DxCruziV2 and its performance as the antibody capture molecule for in-house ELISAs. Panel (**A**) depicts three views of the projected tertiary structure from the I-Tasser server, with the inserted epitopes in red and the core structure in green. Panel (**B**) shows the absorbance values for patient samples previously diagnosed by LACENs in the states of Maranhão, Sergipe, Ceará, and Paraíba in Brazil as positive (green diamonds; n = 60) and negative (red dots; n = 75) for Chagas disease. Panel (**C**) shows the reactivity indexes of healthy individuals (n = 24) and patients with inactive cutaneous Leishmaniasis (Leish-C; n = 1), visceral Leishmaniasis (Leish-V; n = 32), malaria (n = 12), or dengue (n = 20) calculated with the cutoff value obtained from an ROC analysis of the data in Panel (**B**).

**Figure 5 vaccines-12-01029-f005:**
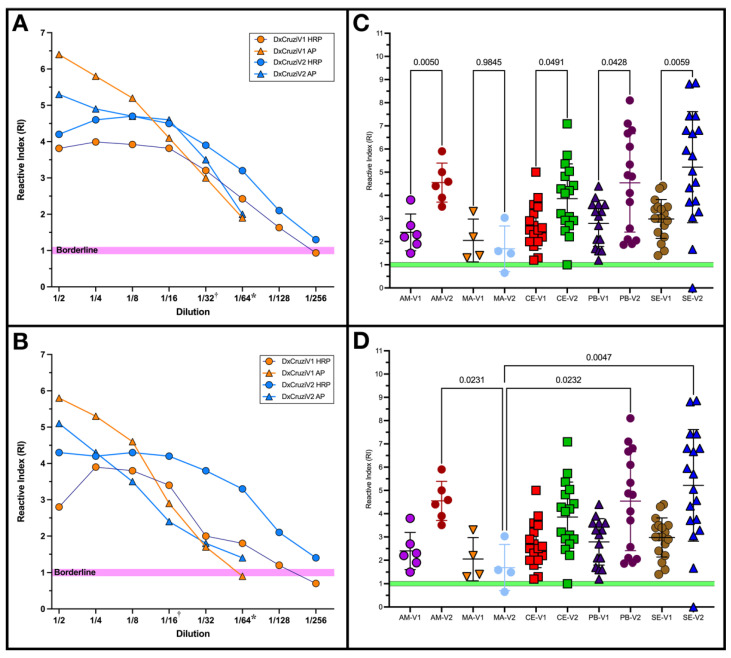
Analytical and geographical sensitivity of DxCruziV1 and DxCruziV2. In-house ELISAs were prepared with 500 ng/well of DxCruziV1 or DxCruziV2. Reactive indices for a 1:2 serial dilution up to 1:256 of the WHO International Biological Standard for Chagas disease 09/188 (Panel (**A**)) and 09/186 (Panel (**B**)) detected using anti-human IgG secondaries antibodies conjugated with horseradish peroxidase (HRP; orange symbols) or alkaline phosphatase (AP; blue symbols). Each data point represents the median of three independent measurements, except for DxCruziV2 HRP at 1:8 dilution of 09/186, which showed zero reactivity in one measurement and was excluded. The influence of the geographical origins of patient samples on performance was evaluated via Brown–Forsythe and Welch ANOVA tests between DxCruziV1 and DxCruziV2 (Panel (**C**)) or DxCruziV2 against different locations (Panel (**D**)) with the *p*-value displayed. Comparisons displayed were insignificant. * The target dilution factor suggested by the WHO [32]. ^†^ Maximum dilution factor with a reactive index > 1 determined via commercial tests in Brazil from a previous study [49]. AM—Amazonas; MA—Maranhão; CE—Ceará; PB—Paraíba; SE—Sergipe.

**Table 1 vaccines-12-01029-t001:** Epitopes utilized to build DxCruziV1 and DxCruziV2 with their insertion site in the ß-receptacle and *T. cruzi* protein origins.

	Epitope	Sequence	Insertion Site ^a^	AA Duplication ^b^	Protein Origin	Refs.
**DxCruzV1**	1	KFAELLEQQKNAQFPGK	20	-	KMP11	[34]
	2	KAAAAPA ^c^	74	KL	TcE	[35,36]
	3	KAAIAPA ^d^	124	DV	TcE	[35,36]
	4	GDKPSPFGQAAAADK	139	GT	PEP-2; B13; Ag2	[37,38,39]
	5	KQKAAEATK	161	LK	CRA	[40,41,42]
	6	AEPKPAEPKS	178	TAD	TcD-2	[36,39]
	7	AEPKSAEPKP	199	GS	TcD-1	[36,39]
	8	GTSEEGSRGGSSMPS	215	DGP	TcLo 1.2	[36]
	9	SPFGQAAAGDK	238	EL	PEP-2; B13; Ag2	[37,38,39]
	10	KQRAAEATK	C-term	-	CRA	[40,41,42]
**DxCruziV2**	1	KFAELLEQQKNAQFPGK	1	-	KMP11	[34]
	11	DSSAHSTPSTPA	74	KL	SAPA	[43,44]
	4	GDKPSPFGQAAAADK	139	GT	PEP-2; B13; Ag2	[37,38,39]
	12	FGQAAAGDKPS	162	KLK	3973 (TcCA-2)	[45]
	6	AEPKPAEPKS	178	TAD	TcD-2	[36]
	13	PPSGTENNKPATG	199	GSGTSWKGS	TSSA	[46]
	8	GTSEEGSRGGSSMPS	215	DGP	TcLo1.2	[36]
	9	SPFGQAAAGDK	238	EL	PEP-2; B13; Ag2	[37,38,39]
	3	KAAIAPA	C-term	GTSWKGD	TcE	[35,36]
	10	KQRAAEATK	C-term	-	CRA	[40,41,42]

^a^. Amino acid numbers are based on the coding sequence of the ß-receptacle in Figure 1A. ^b^. Insertion into a restriction site leads to the duplication of the site and its coding sequence. ^c^. Bold denotes epitopes changed between DxCruziV1 and DxCruziV2. ^d^. Epitope moved to the carboxy terminus in DxCruziV2.

**Table 2 vaccines-12-01029-t002:** Performances of DxCruziV1 and DxCruziV2 at two different cutoff values.

Parameter	DxCruziV1		DxCruziV2	
O.D. 405	>0.2635	>0.3170	>0.2600	>0.3065
Sensitivity (95% CI ^a^)	100.0% (93.98 to 100.0%)	100.0% (93.98 to 100.0%)	96.67% (88.64 to 99.41%)	96.67% (88.64 to 99.41%)
Specificity (95%CI)	98.68% (93.98 to 100.0%)	100.0% (95.19 to 100.0%)	98.68% (92.92 to 99.93%)	100.0% (95.19 to 100.0%)
Kappa Index	0.97	1.00	0.94	0.96
Likelihood Ratio	76.0	-	73.47	-
False Positive	2.7%	0.0%	1.3%	1.3%
False Negative	0.0%	0.0%	3.3%	5.0%
PPV ^b^	96.8	100.0	98.3	98.3
NPV ^c^	100.0	100.0	97.4	96.1
Concordance	98.5%	100.0%	97.8%	97.0%
Non-Concordance	1.5%	0.0	2.2%	3.0%
Area Under Curve	1		0.998	
Std. Error	0		0.001661	
95% CI	1.00 to 1.00		0.995 to 1.00	
*p* value	<0.0001		<0.0001	

^a^. Confidence interval. ^b^. Positive predictive value. ^c^. Negative predictive value.

## Data Availability

The authors declare that the research was conducted in the absence of any commercial or financial relationships that could be construed as potential conflicts of interest.

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
