# Peer review of "Chagas Disease Diagnosis with Trypanosoma cruzi-Exclusive Epitopes in GFP"

_vaccines, 2024, doi:10.3390/vaccines12091029_

Round 1

Reviewer 1 Report

Comments and Suggestions for Authors

Please make all correction as attachment  

Comments on the Quality of English Language

Please make all correction as attachment

Author Response

Rewrite the abstract again to illustrate the aim of this study and write the methods used and clarify the results which the authors reach to it.

Author Response: We agreed that the abstract could be improved and has been replaced with:

“Serological tests are critical tools in the fight against infectious disease. They detect antibodies made during an adaptive immune response against a pathogen with an immunological reagent, whose antibody binding characteristics define the specificity and sensitivity of the assay. While pathogen proteins have conveniently served as reagents, their performance is limited by the natural grouping of specific and non-specific antibody binding sites, epitopes. An attractive solution is to build synthetic proteins that only contains pathogen specific epitopes, which could theoretically reach 100% specificity. However, the genesis of de novo proteins remains a challenge. To address the uncertainty of producing a synthetic protein, we have repurposed the beta barrel of fluorescent proteins into a receptacle that can receive several epitope sequences without compromising its ability to be expressed. Here, two versions of a multi-epitope protein were built using the receptacle that differ by their grouping of epitopes specific to the parasite Trypanosoma cruzi, the causitive agent for Chagas disease. An evaluation of their performance as the capture reagent in ELISAs showed high agreement with recommended diagnostic protocols in a single asssy and suggests that this approach could be applied to other infectious agents.”

Illustrate the serological method more than it -

Author response: We were unsure as to what action the reviewer wanted us to take.

  1. Clarify the number of patient mans infected with chronic Chagas disease and the negative control.

Author response: We have made sure to include in the text references to the number of patient samples used in the evaluation. (Lines 278-279, 283).

  1. The authors clarify the best method for the diagnosis of Chagas disease. IFI or ELISA.

Author Response: There really is no best method, yet. With this manuscript, we argue that an ELISA using our multiepitope protein would be sufficient. Today, the antibody capture molecules in both IFI and commercial ELISAs are full of non-specific epitopes that lead to false positives.   

We have incorporated all of the suggestions below, thank you.

At line 27 generate convert to generating

At line 373 unmeet correct to unmet

At line 379 convert third confirmational to third confirmation test

At line 389 convert contain an to contains an

At line 399 delete were

At line 399convert to those to that of

At line 408 reliably produce to producing proteins

At line 419 of signal to of a signal

At line 423 -424 clearly higher than to clearly higher than that of

At line 424 easily met to meets

Reviewer 2 Report

Comments and Suggestions for Authors

The manuscript by Durans and collaborators is a very elegant and fruitful piece of research, which advances the diagnostics field of Chagas Disease. The insertion of ten different epitopes into the scaffold of the GFP is a clever manner of maximising the exposure of antigens to circulating antibodies. The proof of concept presented in this manuscript is solid, even if the fact that there is not intrinsic fluorescence might imply that the GFP scaffold is not retained.

The work should be completed in the future with a thorough cross-reactivity series of experiments, given the conservation of the epitopes among the trypanosomatids, as shown in the supplementary table S4.

Minor points:

line 84: use "exogenous" better than "extraneous" to describe the sequences

line 132: 100 mM NaCl, instead of Mm

lines 135 and 137: "minutes" instead of "minutos"

line 138: "chromatographic system"

line 139: "omitted"

line 324: please, complete the caption of Table 1

Author Response

Author response: We appreciate the kind words and recognition by the reviewer. Each of the minor points have been addressed and reflects the careful attention by the reviewer. Thank you.

Minor points:

line 84: use "exogenous" better than "extraneous" to describe the sequences

Author response: This was an excellent suggestion. Thank you.

line 132: 100 mM NaCl, instead of Mm

Author response: Thank you for finding this typo.

lines 135 and 137: "minutes" instead of "minutos"

Author response: I try to get my post doctorates to write directly in English but there is usually hints of Portuguese in all documents.

line 138: "chromatographic system"

Author response: Thank you for the suggestion.

line 139: "omitted"

Author response: After seeing this misspelled word, spell checker was reapplied. Thank you.

line 324: please, complete the caption of Table 1

Author response: I returned to the original document and the caption is there. It must have been lost during formating. Thank you.

Reviewer 3 Report

Comments and Suggestions for Authors

This paper discusses the development of synthetic proteins containing specific epitopes for diagnosing Chagas disease. Two multi-epitope proteins, DxCruziV1 and DxCruziV2, were constructed and tested using ELISA. The proteins demonstrated high specificity and sensitivity, offering a reliable method for early and accurate diagnosis of Chagas disease. This approach aims to improve diagnostic performance, reducing false positives and negatives.

strengths

1) High specificity and sensitivity in diagnosing Chagas disease.

2) Potential to reduce the need for multiple tests and improve early detection.

3) Synthetic proteins provide a controlled and replicable diagnostic tool.

I have a few comments for the authors to address:

1) Figure 1B could be optimized by ESPcript (https://espript.ibcp.fr/ESPript/ESPript/), which can show the corresponding secondary structures for the inserted epitope sequences.

2) From the comparison in Table 2 (despite the mislabel as Table 1), the authors should add some details and descriptions of the table to explain why DxCruziV1 is better. In addition, a concise conclusion should be included in the corresponding result section 3.3.

3) Regarding Figure 3, the legends are difficult to understand. Figure 3A contains 24 spots in each row; how are these linked to the “library of consecutive peptides of 14 residues with a six amino acid overlap”? Additionally, why does Figure 3B only contain data from the first 16 samples rather than all 24 samples in Figure 3A?

4) Figure 5B, why does 1/8 dilution have a lower RI than 1/16 dilution for DxCruziV2 HRP?

Others:

Figure S2C please add corresponding legends to explain the samples of “6C-CE…”

Line 274 where is “Table II”? or “Table 2”?

Line 278  missing “DxCruziV1” before “and”; In addition, I guess this is Table 2?

In Table 1, “Kappa Index” should be explained in the materials and methods section.

Line 296-297 please indicate the three specific residues or spots in Figure 3A that could not capture anti-TZ antibody.

Line 342 “Panel A” and “Panel B” should be “Figure 5A” and “Figure 5B”

Author Response

1) Figure 1B could be optimized by ESPcript (https://espript.ibcp.fr/ESPript/ESPript/), which can show the corresponding secondary structures for the inserted epitope sequences.

Author response: This was an excellent suggestion. However, when we examined the results of the analysis, the program was only able to recognize secondary structure in a few regions and not across the whole coding sequence. This could be due to the interuptions caused by inserting sequences or our inexperience with the program. We have resolved to maintain the figure as is, but will be checking its application in the future. Thank you.

2) From the comparison in Table 2 (despite the mislabel as Table 1), the authors should add some details and descriptions of the table to explain why DxCruziV1 is better. In addition, a concise conclusion should be included in the corresponding result section 3.3.

Author response: In fact, we believe that V2 is better based on the clinical sensitivity with the WHO Standards and the geographical performance profile that showed greater reactivity in most regions. While V1 does show an excellent performance, we think that there is some variablity in the preparation of protein rather than lower performance.

3) Regarding Figure 3, the legends are difficult to understand. Figure 3A contains 24 spots in each row; how are these linked to the “library of consecutive peptides of 14 residues with a six amino acid overlap”? Additionally, why does Figure 3B only contain data from the first 16 samples rather than all 24 samples in Figure 3A?

Author response: We have rewritten the legend to make it clear that the coding sequence was synthesized in duplicate in a grid pattern of three rows and twenty-four across (A1-A24, B1-B24, C1-C16) and (D1-D24, E1-E24, F1-F16). Lines 315-318.

4) Figure 5B, why does 1/8 dilution have a lower RI than 1/16 dilution for DxCruziV2 HRP?

Author response: We really appreciate the reviewer for calling out this data point. As each is the median of three measurements, we returned to the original data and it was clear that one of the three wells did not have any signal. Zero. Therefore, we removed the data point from the graph and mention it in the legend.

Lines 368-370: Each data point represents the median of three independent measurements except DxCruziV2 HRP at 1:8 dilution that showed zero reactivity in one measurement.

Figure S2C please add corresponding legends to explain the samples of “6C-CE…”

Author response: The significance of the nomenclature was included in Lines 512-513.

Line 274 where is “Table II”? or “Table 2”?

Line 278  missing “DxCruziV1” before “and”; In addition, I guess this is Table 2?

Author response: There was some mixup between versions that have been corrected.

In Table 1, “Kappa Index” should be explained in the materials and methods section.

Author response: A descriptor was included in Lines 202-203.

Line 296-297 please indicate the three specific residues or spots in Figure 3A that could not capture anti-TZ antibody.

Author response: We have referred to the specific epitope number as well as their sequence in Table 1.

Line 342 “Panel A” and “Panel B” should be “Figure 5A” and “Figure 5B”

Author response: We have changed the text.

Round 2

Reviewer 3 Report

Comments and Suggestions for Authors

The authors have addessed most of my concerns and this paper is ready to go!